# In Vitro and In Silico Screening of Anti-*Vibrio* spp., Antibiofilm, Antioxidant and Anti-Quorum Sensing Activities of *Cuminum cyminum* L. Volatile Oil

**DOI:** 10.3390/plants11172236

**Published:** 2022-08-29

**Authors:** Siwar Ghannay, Kaïss Aouadi, Adel Kadri, Mejdi Snoussi

**Affiliations:** 1Department of Chemistry, College of Science, Qassim University, Buraidah 51452, Saudi Arabia; 2Faculty of Sciences of Monastir, University of Monastir, Avenue of the Environment, Monastir 5019, Tunisia; 3Faculty of Science of Sfax, Department of Chemistry, University of Sfax, B.P. 1171, Sfax 3000, Tunisia; 4Faculty of Science and Arts in Baljurashi, Albaha University, P.O. Box 1988, Albaha 65527, Saudi Arabia; 5Department of Biology, College of Science, Hail University, P.O. Box 2440, Ha’il 2440, Saudi Arabia; 6Laboratory of Genetics, Biodiversity and Valorization of Bio-Resources (LR11ES41), Higher Institute of Biotechnology of Monastir, University of Monastir, Avenue Tahar Haddad, BP74, Monastir 5000, Tunisia

**Keywords:** *Cuminum cyminum* L., phytochemistry, *Vibrio* spp., antioxidant, in silico approach

## Abstract

*Cuminum cyminum* L. essential oil (cumin EO) was studied for its chemical composition, antioxidant and vibriocidal activities. Inhibition of biofilm formation and secretion of some virulence properties controlled by the quorum sensing system in *Chromobacterium violaceum* and *Pseudomonas aeruginosa* strains were also reported. The obtained results showed that cuminaldehyde (44.2%) was the dominant compound followed by β-pinene (15.1%), γ-terpinene (14.4%), and *p*-cymene (14.2%). Using the disc diffusion assay, cumin EO (10 mg/disc) was particularly active against all fifteen *Vibrio* species, and the highest diameter of growth inhibition zone was recorded against *Vibrio fluvialis* (41.33 ± 1.15 mm), *Vibrio parahaemolyticus* (39.67 ± 0.58 mm), and *Vibrio natrigens* (36.67 ± 0.58 mm). At low concentration (MICs value from 0.023–0.046 mg/mL), cumin EO inhibited the growth of all *Vibrio* strains, and concentrations as low as 1.5 mg/mL were necessary to kill them (MBCs values from 1.5–12 mg/mL). Using four antioxidant assays, cumin EO exhibited a good result as compared to standard molecules (DPPH = 8 ± 0.54 mg/mL; reducing power = 3.5 ± 0.38 mg/mL; β-carotene = 3.8 ± 0.34 mg/mL; chelating power = 8.4 ± 0.14 mg/mL). More interestingly, at 2x MIC value, cumin EO inhibited the formation of biofilm by *Vibrio alginolyticus* (9.96 ± 1%), *V. parahaemolyticus* (15.45 ± 0.7%), *Vibrio cholerae* (14.9 ± 0.4%), and *Vibrio vulnificus* (18.14 ± 0.3%). In addition, cumin EO and cuminaldehyde inhibited the production of violacein on Lauria Bertani medium (19 mm and 35 mm, respectively). Meanwhile, 50% of violacein inhibition concentration (VIC_50%_) was about 2.746 mg/mL for cumin EO and 1.676 mg/mL for cuminaldehyde. Moreover, elastase and protease production and flagellar motility in *P. aeruginosa* were inhibited at low concentrations of cumin EO and cuminaldehyde. The adopted in-silico approach revealed good ADMET properties as well as a high binding score of the main compounds with target proteins (1JIJ, 2UV0, 1HD2, and 3QP1). Overall, the obtained results highlighted the effectiveness of cumin EO to prevent spoilage with *Vibrio* species and to interfere with the quorum sensing system in Gram-negative bacteria by inhibiting the flagellar motility, formation of biofilm, and the secretion of some virulence enzymes.

## 1. Introduction

Infectious diseases, reinforced by the emergence of antibiotic resistant pathogens are known as a high leading cause of death in the world lead causing higher mortality and morbidity and increased healthcare costs [1,2]. Antimicrobial resistance (AMR) represents the acquired ability of pathogens to withstand antimicrobial treatment is an increasing global concern results from the abuse and misuse of antibiotics have been recognized as one of the top health threats to human society [3]. A recent study revealed that microorganisms responsible for various human infections (~80%) and hospital-acquired infections (60–70%), have shown a biofilm origin [4]. Biofilms as a cellular conformation confers survival properties to microbial populations which are attached to a surface, enveloped and organized in an exopolysaccharide matrix, play an important role in the development of antimicrobial resistance [5]. Many genes and environmental factors were implicated in the formation of biofilm by *P. aeruginosa* strains known for their high drug resistance against traditional antibiotic therapy [6,7]. In fact, *P. aeruginosa* is a human pathogen that is frequently responsible for hospital-acquired infections and is the main cause of morbidity and mortality in cystic fibrosis patients [8]. In *P. aeruginosa*, LasR and RhlR are homologous LuxR-type soluble transcription factor receptors that bind their cognate AIs and activate the expression of genes encoding functions required for virulence and biofilm formation [9]. To eradicate the problem of biofilm formation, the QS inhibitory activity remains a significant strategy. Aromatic and medicinal plants represent a rich source of novel lead compounds that have been traditionally used in phytotherapy [10,11]. Herbs, spices and derived extracts are gaining more popularity and have been used for treating several disorders and diseases due to the inherent medicinal properties, due to their antioxidant [12,13,14,15,16,17,18,19], antibacterial [16], anti-inflammatory [16], antimicrobial [17,18,19,20,21], wound healing [20], cytotoxicity [20], anti-acetylcholinesterase [21,22], and antidiabetic [22] potential. They have been largely used in food and beverages to enhance flavor, aroma and color [23,24,25].

*Cuminum cyminum* L., known as “KAMMOUN” is a member of the Apiaceae (Umbelliferae) family, just like parsley. Cumin is an annual, herbaceous, medicinal spice and culinary plant (15 to 50 cm high) [26]. The plant is largely cultivated in arid and semi-arid areas, including India, Middle East, China and Mediterranean region [27]. The stems are hollow and grooved, with alternate leaves, digested, light green, without stipules. The small, white flowers have five petals, in umbels. The seeds are long, straight brown, longitudinal ribs of 5–6 mm. Appear in pairs on the branches. As a condiment, cumin is extensively used as food additive and flavoring agent in different cuisines, essentially in South Asian, Northern African, and Latin American cuisines [28]. The nutritional values and health benefits of cumin seeds have been reported demonstrating their uses in the treatment of fever, flatulence, loss of appetite, wounds, diarrhea, vomiting, abdominal distension, edema and puerperal disorders as well as increase the appetite, taste perception, digestion, vision, strength and lactation [27]. The pharmacological activities of this plant have been reported, revealing the ability of this plant to exert antimicrobial, insecticidal, anti-inflammatory, analgesic, antioxidant, anticancer, antidiabetic, antiplatelet aggregation, hypotensive, bronchodilators, immunological, contraceptive, anti-amyloidogenic, anti-osteoporotic, protective, and central nervous effects [29]. Phytochemical’s analysis of cumin showed that was a reach sources of coumarin, flavonoid, anthraquinone, alkaloid, glycoside, protein, resin, saponin, tannin and steroid [30].

Hence, in view of the attributed medicinal significance of the cumin plant and its availability as medicinally resource, the present work focuses specifically on an aromatic plant commonly used in the Saudi kitchen to prepare fish and shellfish dishes. We aimed to explore the constituents of its EO and to evaluate in vitro, its anti-*Vibrio* activities. The ability of the obtained cumin EO to scavenge reactive oxygen species using different assays was also assessed. Moreover, a computational study has been performed to elucidate the physicochemical properties, pharmacokinetic properties, druglikeness, and toxicity prediction of the main bioactive compounds from cumin EO. To get insight into the interaction mode of these bioactive molecules with known target enzymes involved in antioxidant, antibacterial, and anti-quorum sensing activities, a molecular docking approach was adopted.

## 2. Results

### 2.1. Phytochemical Composition

Table 1 summarized the phytochemical composition of cumin EO obtained by hydrodistillation technique of seeds. Twenty chemical compounds were identified representing 99.1% of the total identified phytoconstituents. This volatile oil was dominated by oxygenated monoterpenes (51.3%) and monoterpene hydrocarbons (46.7%). The main compounds identified in cumin EO were cuminaldehyde (42.4%), β-pinene (15.1%), γ-terpinene (14.4%), *p*-cymene (14.2%), and α-terpin-7-al (5.2%).

### 2.2. Antioxidant Activities

Table 2 summarizes the results of the antioxidant activities of cumin EO as compared to well-known standard molecules evaluated by using DPPH, reducing power, β-carotene, and chelating power assays. The obtained results reveal promising antioxidant activities at low concentrations as compared to ascorbic acid (AA), butylated hydroxytoluene (BHT), and butylate hydroxyanisole (BHA). In fact, IC_50_ for the DPPH test was about 8 ± 0.54 mg/mL, 3.8 ± 0.34 mg/mL for the β-carotene test, and 8.4 ± 0.14 mg/mL for the chelating power test. 

### 2.3. Antimicrobial Activity

The ability of the obtained cumin EO was tested against fifteen ***Vibrio*** species. Results revealed a bacteriostatic action of the tested oil (MBC/MIC ratio > 4). The growth of almost all *Vibrio* species on liquid media was inhibited at low concentrations ranging from 0.023 to 0.046 mg/mL. In addition, the same bacteria were completely killed by low concentration of cumin EO varying from 1.5 to 12 mg/mL. The mean diameter of growth inhibition zone obtained by the disc diffusion agar test at 10mg/disc confirms the high activity of cumin EO against almost all *Vibrio* species with mean diameter of inhibition zone (mZI) of approximately 34.33 ± 0.58 mm for *V. cholerae* ATCC 9459, 39.67 ± 0.58 mm for *V. parahaemolyticus* ATCC 17802, 34.67 ± 0.58 mm for *V. alginolyticus* ATCC 33787, and 30.33 ± 0.58 mm for *V. vulnificus* ATCC 27562. All results are summarized in Table 3.

### 2.4. Biofilm Inhibition 

Cumin EO was tested for its ability to inhibit the biofilm formation on polystyrene 96 well-plate by four *Vibrio* species including *V. cholerae*, *V. vulnificus*, *V. parahaemolyticus*, and *V. alginolyticus* by using XTT technique. Results showed that the examined oil was able to inhibit the biofilm formation of the tested *Vibrio* species in a concentration-dependent manner. In fact, at 2xMIC, the inhibition was about 9.96 ± 1.00% against *V. alginolyticus* ATCC 33787) and 18.14 ± 0.30% against *V. cholerae* ATCC 9459. Interestingly, at 50 mg/mL, the highest percentage of biofilm formation inhibition was recorded for all strains reaching a percentage between 66.29 ± 3% (*V. cholerae* ATCC 9459) and 76.29 ± 4%. All these data are summarized in Figure 1.

### 2.5. Anti-QS Activity

#### 2.5.1. Qualitative and Quantitative Violacein Inhibition Estimation

The ability of cumin EO and its major compound (cuminaldehyde) to inhibit the production of violacein by *C. violaceum* CV026 was tested at 2 mg/mL (Figure 2). The inhibition zone of the EO was about 32 mm and about 35 mm for its main compound (cuminaldehyde). Meanwhile, the anti-QS sensing zone of cuminaldehyde was interestingly higher than the EO (35 mm and 19 mm, respectively).

More interestingly, quantitative estimation on the effect of various concentration of cumin volatile oil on the growth of *C. violaceum*, showed a MIC value about 5 mg/mL and the VIC_50%_ was about 2.746 mg/mL. Meanwhile, for the main compound (Cuminaldehyde), MIC and VIC_50%_ values were about 1.25 mg/mL about 1.676 mg/mL, respectively. 

#### 2.5.2. Anti-Swarming Activity

The starter strain (*P. aeruginosa* PAO1) was used to test the effect of cumin EO and cumin aldehyde at different concentrations on its motility on semi-solid agar plates. The results obtained are summarized in Table 4. At 10 mm/mL, the motility of this bacterium was more inhibited by cuminaldehyde (by 70.99 ± 0.57%) as compared to the EO (64.20 ± 0.57%). At higher concentration (500 mg/mL), the percentage of motility inhibition was about 89.77 ± 0.00% for cuminaldehyde and 90.12 ± 0.57% for the cumin EO.

#### 2.5.3. Elastase and Protease Inhibition

*Pseudomonas aeruginosa* is able to produce several virulence factors responsible for its pathogenecity like alkaline proteases, elastases, and collagenase. Our results showed that the obtained cumin EO and its main compounds are able to modulate the production of elastase and protease with different degree and in a concentration dependent manner (Figure 3). In fact, cumin EO and cuminaldehyde decreased the production of protease by 68.32% and 71.09% respectively at 0.05 mg/mL. Similarly, at high concentration (2.5 mg/mL), cumin EO inhibited the production of protease by 82.14%, and by 83.43% for cuminaldehyde. More interestingly, cumin EO inhibited the production of elastase by 46.08% for cuminaldehyde and by 43.34% for the volatile oil. At 2.5 mg/mL, elastase production in *P. aeruginosa* PAO1 was inhibited by 63.14% and 62.12% respectively for cumin EO and its main compound (cuminaldehyde).

### 2.6. ADMET Analysis

The in silico ADMET prediction of the selected major compounds (Table 5) revealed a good permeability on intestinal Caco-2 cells and is easy to be absorbed, with values in the range of 1.373–1.517, and high intestinal human absorption (above 94%), with only 16 and 17 exhibited low skin permeability. All phytocompounds were expected to not act on P-glycoprotein, are likely to cross the blood-brain barrier (BBB) with 9, 16 and 17 are able to slightly access to the central nervous system (CNS). Another important parameter used in distribution named distribution volume which characterize the distribution of drugs in various tissues in vivo. Predictive data showed that compound 4 was well distributed, 7 and 9 were moderately, but 16 and 17 were relatively lower distributed.

Cytochrome P450s is an important enzyme system for drug metabolism in liver, with the most important where subtypes are CYP2D6 and CYP3A4. Results indicate that none of the selected compounds will be metabolized by the cytochrome P450s enzymes. Regarding toxicity parameters, our phytocompounds may not inhibit the hERG channel and have no AMES nor hepatotoxicity profile.

### 2.7. Molecular Docking Analysis

In order to assess the potential of cumin EO to inhibit the growth of pathogenic microorganisms and to reduce hydrogen peroxide and alkyl hydroperoxides, molecular docking study was performed to gain insight into the most preferred binding mode of compound into the enzyme binding active site. Ligands have been selected based on their abundance in the EO (%) and their lowest binding score.

*Staphylococcus aureus* tyrosyl-tRNA synthetase (PDB ID, 1JIJ): inhibitors of tyrosyl-tRNA synthetase could be promising drug candidates leading to high selectivity and broad-spectrum antibacterial agents. As shown in Table 6 and Figure 4, cuminaldehyde form C-H bond: Gly192 (2.81). Alkyl/Pi-Alkyl: Cys37 (5.21), Leu70 (4.89) (5.39), however β-Caryophyllene was involved via Alkyl/Pi-Alkyl: Cys37(4.64), Ala39 (4.28) (4.53) (4.75), Pro53 (5.37) (4.50), His50 (4.00) (5.06) with *S. aureus* tyrosyl-tRNA synthetase.

Human peroxiredoxin 5 (PRDX5) receptor (PDB ID, 1HD2) is a potential target for the evaluation of antioxidant activity which permits the reduction of hydrogen peroxide and alkyl peroxide, with the help of thiol-containing donor molecules. The major and the most relevant docked phytocompounds were β-pinene which interact preferentially via Alkyl/Pi-Alkyl with Pro40 (4.05) (4.38), Pro45 (5.05), Cys47 (4.99), Leu116 (5.11), Phe120 (4.88) residues. On the other hand, cuminaldehyde interact with Thr147 (2.10). Alkyl: Pro45 (5.14), Cys47 (5.00) residues by H bond interactions (Table 5 and Figure 5).

LasR enzyme (PDB ID, 2UV0) and CviR enzyme (PDB ID, 3QP1): To combat multidrug resistant bacteria, QS inhibition strategies remains a promising strategy due to their ability to regulate pathogenicity and virulence. For this, docking studies were performed towards two target QS receptors, LasR enzyme (PDB ID, 2UV0) and CviR enzyme (PDB ID, 3QP1) able of inhibiting *P. aeruginosa* bacterium. CviR is receptor protein of *C. violaceum* 12472 and LasR is transcriptional activator of *P. aeruginosa* virulence factors. All the selected compounds were able to bind in the evaluated structures of the CviR and LasR with the following binding scores and binding residues (Table 4).

The best selected bioactive phytocompounds in *C. cyminum* L. EO with LasR enzyme were *p*-cymene (Figure 6A) which was able to bind via the following interactions: van der Waals with Leu110, Unfavorable Bump with Trp88 (0.69) (1.13) (1.47) (1.49), Pi-Pi T-Shaped with Tyr56 (5.01) and Alkyl/Pi-Alkyl with Leu36 (4.93), Trp88 (4.87).g-Terpinene form Unfavorable Bump with Trp88 (0.61) (1.34) (1.54) and Alkyl/Pi-Alkyl with Leu36 (4.93), Tyr56 (5.20), Tyr64 (3.76) (4.88), Trp88 (4.67). However, cuminaldehyde involved H bond (Arg61 (4.31)), Pi-Lone Pair (Tyr64 (2.79)), Unfavorable Bump (Trp88 (0.71) (0.75) (0.22) (1.39)), Pi-Pi T-Shaped (Tyr56 (4.90)), and Alkyl/Pi-Alkyl (Trp88 (4.75)) interactions. 

The selected phytocompounds with CviR enzyme were p-Cymene, forming the CviR-p-cymene complex (−7.5 kcal/mol), which was stabilized by the following interactions (Figure 6B): Unfavorable Bump with Trp111 (1.25) (1.58), Pi-Pi T-Shaped with Tyr80 (5.80), Alkyl/Pi-Alkyl with Trp84 (3.37) Ile99 (4.80), Phe126m(4.94), Ala130 (4.88), Met135 (5.80) residues. The complex CviR-g-Terpinene (−7.5 kcal/mol) form van der Waals: Leu57, trp84, Tyr88, Ile99, Leu100. Unfavorable Bump: Trp111 (0.90) (1.41). Alkyl/Pi-Alkyl: Tyr80 (5.36), Phe115 (4.99), Phe126 (4.99), Ala130 (5.01), Met135 (3.80) (5.35), Trp111 (4.20) (4.45), however, CviR-cuminaldehyde (−7.5 kcal/mol) form Pi-Pi T-Shaped with Tyr80 (5.82) and Pi-Alkyl with Ile99 (4.76) residues 

Table 6 summarizes the best obtained poses based on the binding energy with the dominant compounds.

## 3. Discussion

Cumin seeds are largely used as a flavoring and food preservative agent due to their richness in bio-compounds with a large array of biological activities [5]. 

In this study, the volatile oil extracted from cumin seeds by hydrodistillation is a rich source of cuminaldehyde (42.4%). In fact, it is well documented that the chemical composition of cumin seeds depends on several endogenous (cultivar, genetic traits) and exogenous factors (geographical region, harvesting time, and extraction procedures). Different percentages of cuminaldehyde were reported from cumin seeds around the word as summarized in Table 7.

Our results revealed that the obtained EO was active against fifteen *Vibrio* species with different degrees. The diameter of growth inhibition zone ranged from 11 ± 00 mm (*V. diazotrophicus* ATCC 33466) to 41.33 ± 1.15 mm (*V. fluvialis* ATCC 33809). Cumin EO oil exhibited bacteriostatic activity against all *Vibrio* species with MICs and MBCs values ranging from 0.023–0.046 mg/mL and 1.5–12 mg/mL, respectively. Our results are in agreement with previous study who demonstrated that cumin EO is active against a wide spectrum of microorganisms [32,33,47]. More recently, it has been reported that cumin EO from Iran (cuminaldehyde 38.26%) was active against multidrug resistant *Staphylococcus aureus* (*S. aureus*) strains with MICs and MBCs values ranging from 5 to 10 and 10 to 20 µL/mL, respectively [38]. 

This antimicrobial activity can be positively correlated with the concentration of aldehydes (cuminaldehyde) and terpene group (mainly α-pinene and β-pinene). Using cumin EO (Cuminaldehyde 39.78%), Hajlaoui and colleagues [47] have demonstrated a large antimicrobial activity against Gram-positive bacteria (*S. aureus*, *S. epidermidis*, *Micrococcus luteus*, *Bacillus cereus*), Gram negative bacteria (*Escherichia coli*, *Enterococcus faecalis*, *P. aeruginosa*, *Salmonella typhimirium*, *Listeria monocytogenes*), twelve *Vibrio* species, and yeast strains (*Candida albicans, Candida tropicalis*, *Candida glabrata*, *Saccharomyces cerevisiae*). More recently, our team demonstrated that caraway (*Carum carvi* L.) EO was active against the same *Vibrio* species tested in the present study with a high diameter of growth inhibition zone and low MIC and MBC values [53]. Table 8 represents a systemic review of the bibliography describing the effect of some EOs against different members of Vibrionaceae family.

More interestingly, our EO exhibited antioxidant activities as revealed by DPPH (IC_50_= 8 ± 0.54 mg/mL), reducing power (EC_50_ = 3.5 ± 0.03 mg/mL), β-carotene (IC_50_ = 3.8 ± 0.34 mg/mL), and chelating power (IC_50_ = 8.4 ± 0.14 mg/mL) assays in comparison with BHA, BHT, and ascorbic acid. Previous results have discussed the antioxidant activity of cumin essential oil from different origin [46,48,78,79,80]. 

In addition, our results showed that cumin EO (Chemotype cuminaldehyde) was able to inhibit the biofilm formation of *V. alginolyticus* ATCC 33787, *V. parahaemolyticus* ATCC 17802, *V. vulnificus* ATCC 27962, and *V. cholerae* ATCC 9454 at MICs value ranging from 9.96 to 18.14%. The biofilm formation by these strains was highly inhibited at MBCs values, and at 50 mg/mL. Similar results have reported the effectiveness of EO from *P. crispum*, *O. basilicum*, *M. spicata*, *C. carvi* to inhibit the biofilm formation by the same strains [53,67,68,69]. It has been also demonstrated that clove, garlic, and thyme volatile oils are able to inhibit the formation of biofilm by *V. parahaemolyticus* at 8xMIC (0.56% for clove, 0.16% for thyme, and 0.72% for garlic) after 30 min of application of the volatile oils. Cumin EO was described to inhibit the biofilm formation by clinical *Klebsiella pneumoniae* on semiglass lamellas [81] and the attachment of *E. coli* MTCC 40, *Salmonella* spp. MTCC 1163 and *S. aureus* MTCC 7443 strains on microtiter plate by 52.11% [82]. 

In this work, we evaluated the effect of cumin EO and cuminaldehyde to inhibit the production of violacein by *C. violaceum* using both qualitative and quantitative methods. Previous reports have shown that *C. cyminum* exhibited potent inhibition of violacein production in *C. violaceum* at low concentration (0.5 mg/mL of methanolic extract), as well as swarming and swimming motility in *P. aeruginosa* PAO1 at 60 μg/mL [83]. In addition, methanolic extract of cumin seeds was able to inhibit the violacein production by *C. violaceum,* exopolysaccharide production, flagellar motility, and biofilm formation [84].

Overall, the biological activity of the tested cumin EO (anti-*Vibrio* spp., antioxidant, antibiofilm, and anti-quorum sensing properties) can be explicated by the high percentage of cuminaldehyde (42.4%), β-pinene (15.1%), γ-terpinene (14.4%), and *p*-cymene (14.2%). In fact, cuminaldehyde was described to be active against biofilm-forming *K. pneumoniae* and *P. aeruginosa* strains [81,85]. In addition, this aldehyde was described to be active against planktonic *B. cereus*, *B. licheneformis*, *S. aureus*, *E. coli*, *P. fluorescens*, *P. aeruginosa*, *P. fragi*, *S. paratyphi*, *S. abony*, and *S. Typhi* strains [45,51,85,86]. More recently, it has been reported that cuminaldehyde enhances the antimicrobial potential of ciprofloxacin tested against *S. aureus* and *E. coli* strains [87]. In addition, Chen et al. [78] highlighted the role cuminaldehyde, β-pinene, *p*-cymene, and γ-terpinene as promising scavenging molecules of various reactive oxygen species. 

## 4. Materials and Methods

### 4.1. Plant Material and Extraction Procedure

Cumin seeds (*Cuminum cyminum* L.) were purchased from a local market in August 2021. The taxonomic position was evaluated by Dr. Zouhair Noumi, University of Sfax, Tunisia (Voucher No: AN-0005). The volatile oil was extracted by using hydrodistillation technique [53].

### 4.2. Analysis of the Volatile Compounds

GC/EIMS analyses was performed with a Varian CP-3800 GC equipped with a HP-5 capillary column (30 m × 0.25 mm; coating thickness 0.25 μm) and a Varian Saturn 2000 ion trap mass detector. The identification of compounds was done by comparison of their Kovats retention indices (Ri) [determined relative to the tR of n-alkanes (C10–C35)], with either those of the literature and mass spectra of authentic compounds available in our laboratories by means of NIST 02 and Wiley 275 libraries. The components’ relative concentrations were obtained by peak area normalization [18].

### 4.3. Sceening of the Anti-Vibrio spp. Activity

Fifteen *Vibrio* species (17 bacteria) commonly isolated from aquatic environment ant their associated organisms were used in this study. Semi-quantitative disc diffusion technique on Mueller Hinton-1%NaCl Petri dishes was used to estimate the growth inhibition zone around sterile Whatmann disc impregnated with 10 mg of cumin EO [53,67,68]. For the experiment, *Vibrio* strains were grown on Mueller-Hinton supplemented with 1% NaCl. Fresh Petri dishes were inoculated using bacterial suspension (optical density was adjusted to 0.5 McFarland) by cotton swab technique. Sterile filter paper disks (6 mm in diameter, Biolife, Milan, Italy) were impregnated with 10 mg of cumin EO and then placed on the inoculated Petri dishes. After sitting overnight at 37 °C, the diameter of growth inhibition zone around the disks was estimated using a 1-cm flat ruler.

The determination of the lowest concentration able to inhibit the growth and/or to kill the tested *Vibrio* species was estimated by using microdilution technique as reviously described by Snoussi et al. [58]. In fact, a twofold serial dilution of cumin EO in DMSO-5% was prepared in 96-well plates, starting from 25 µL/mL (23.125 mg/mL), in Mueller-Hinton Broth-1% NaCl. Five microliters of microbial inoculum were added to each well containing 100 µL of the serially diluted volatile oil. After incubation at 37 °C, the minimum inhibitory concentration (MIC) was defined as the lowest concentration able to inhibit the growth of a specific microorganism. To determine the minimum bactericidal concentration (MBC), 3 µL from all the wells with no visible growth were point-inoculated in Mueller-Hinton (1% NaCl) agar. After 24 h of incubation, the concentration at which the *Vibrio* spp. strain presents no growth is recorded as the MBC value.

### 4.4. Evaluation of the Antioxidant Activities

The antioxidant activity experiments were carried out by using four different assays: DPPH, β-Carotene bleaching, and reducing/chelating power assays by using the protocols previously described Ghannay et al. [53].

### 4.5. Inhibition of Virulence Factors Regulated by QS System

#### 4.5.1. Inhibition of Violacein

*Chromobacterium violaceum* (CV026) strain was selected to study the effect of cumin EO against the production of violacein by using disc diffusion assay on LB-agar Petri dishes (2 mm/disc). Twofold serial dilutions of cumin EO were prepared in 96-well plates starting from 5 mg/mL in LB broth and inoculated with *C. violaceum* ATCC 12472 [88].

#### 4.5.2. Biofilm Inhibition

The ability of the tested cumin EO to inhibit the biofilm formation by four *Vibrio* species (*V. alginolyticus*, *V. parahaemolyticus*, *V. vulnificus*, and *V. cholerae*) on a 96 well plate was tested at different concentrations ranging from 2xMIC to 50 mg/mL by using the same protocol described by Ghannay et al. [53].

#### 4.5.3. Effect on Flagellar Motility

*Pseudomonas aeruginosa* PAO1 was used to study the effect of cumin EO at different concentrations on its motility on semi-solid Lauria Bertani (LB-0.3% agar-agar) by using the same protocol described by Snoussi et al. [88].

#### 4.5.4. Elastase and Protease Inhibition in *P. aeruginosa* PAO1

The effect of cumin on the production of elastase by *P. aeruginosa* PAO1 was tested in Elastin Congo Red buffer supplemented with 0.05, 0.5, 0.625, 1.25, and 2.5 mg/mL of the volatile oil. For the protease inhibition, 3 mg of azocasein (Sigma, Tokyo, Japan) was used as enzyme. 

### 4.6. Computational Approach

The receptor proteins (PDB ID: 1HD2, 1JIJ, 2UV0, and 2QP1) were selected from the RSCB protein data bank (http://www.rcsb.org/ accessed on 15 December 2021). Water molecules and co-crystal ligands were removed from each of the protein. AutoGrid was used to create a grid map using a grid box. The grid size and grid dimensions were set for each protein according to the binding pocket are as follow: 1HD2 (Grid size 40 × 40 × 40; Grid dimension center 7.089, 41.659, 34.385; Grid spacing in Å 0.375), 1JIJ (Grid size 40 × 40 × 40; Grid dimension center-11.273, 13.817, 86.080; Grid spacing in Å 0.375), 2UVO (Grid size 440 × 40 × 40; Grid dimension center 23.998, 16.050, 80.315; Grid spacing in Å 0.375), and 3QP1(Grid size 38 × 40 × 40; Grid dimension center 20.546, 12.912, 49.410; Grid spacing in Å 0.375. Docking conditions and steps are previously described by Ghannay et al. [53]. 

### 4.7. ADMET Predicted Properties

The ADMET predictor remains one of the powerful tools for the enhancement of drug design [89,90,91,92,93]. In order to discover effective compounds with better ADMET and drug-likeliness properties, the ADMET profiles of the top major identified compounds were predicted using ADMET SAR online server (http://lmmd.ecust.edu.cn:8000/ accessed on 15 December 2021).

### 4.8. Statistical Analysis

Average values of three replicates were calculated using the SPSS 25.0 statistical package for Windows. Differences in means were calculated using the Duncan’s multiple-range tests for means with a 95% confidence interval (*p* ≤ 0.05).

## 5. Conclusions

In summary, our results indicated that cuminaldehyde, β-pinene, γ-terpinene, and *p*-cymene were the main phytoconstituents identified in cumin EO by GC/MS technique. This chemovar was particularly active against planktonic and biofilm forming *V. alginolyticus*, *V. cholerae*, *V. vulnificus*, and *V. parahaemolyticus* species. The same EO and its main compound (cuminaldehyde) were able to modulate the expression of violacein production in *C. violaceum* in a concentration dependent manner. At low concentrations, cumin EO and cuminaldehyde were able to inhibit the flagellar motility of *P. aeruginosa* PAO1 strain and attenuate the production of elastase and protease. Further analyses are necessary to elucidate the mechanism of action of cumin EO and its role in the prevention of seafood product contamination by spoilage bacteria belonging to *Vibrio* genus.

## Figures and Tables

**Figure 1 plants-11-02236-f001:**
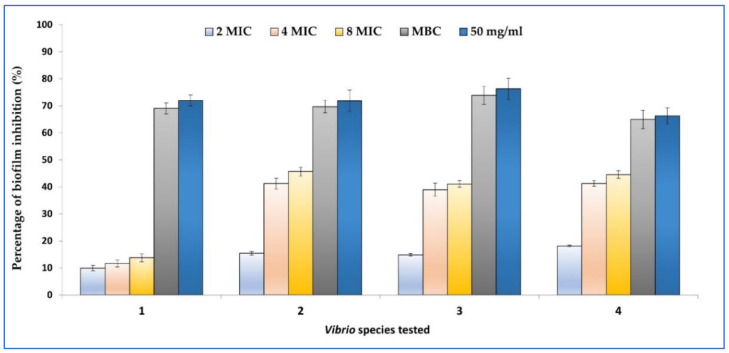
Evaluation of the percentage of biofilm formation inhibition tested by using the colorimetric XTT technique against *V. alginolyticus* ATCC 33787, *V. parahaemolyticus* ATCC 17802, *V. vulnificus* ATCC 27962, and *V. cholerae* ATCC 9459. Errors bars represent standard deviation from three determinations.

**Figure 2 plants-11-02236-f002:**
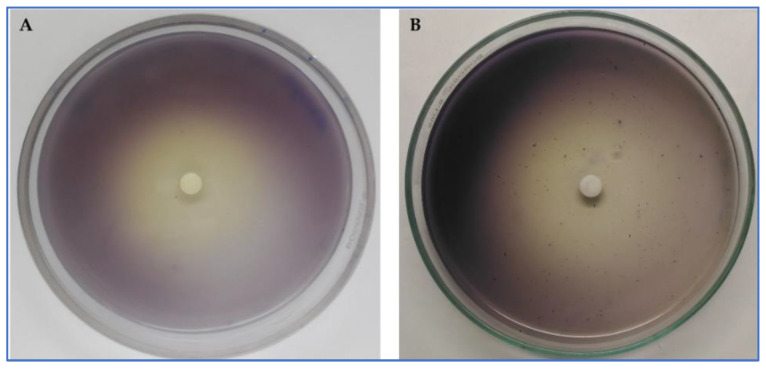
Violacein inhibition by cumin EO (**A**) and its main component (cuminaldehyde, **B**).

**Figure 3 plants-11-02236-f003:**
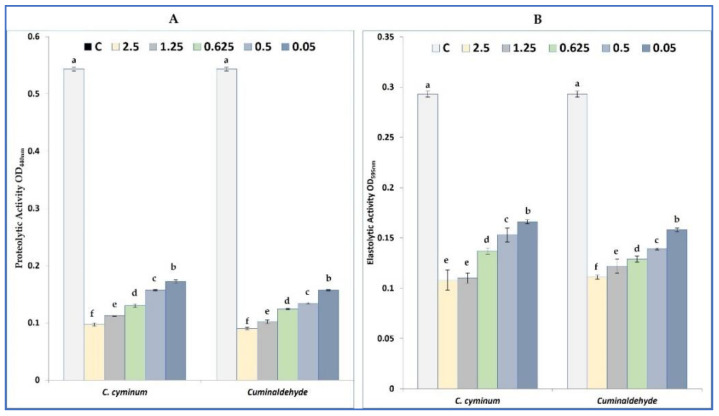
Inhibition of the proteolytic activity (**A**) and elastolytic activity (**B**) in *P. aeruginosa* PAO1 strain by different concentration of cumin EO and cuminaldehyde. Values are the average of at least three independent determinations. Means followed by the same letters are not significantly different at *p* < 0.05 based on Duncan’s multiple range test.

**Figure 4 plants-11-02236-f004:**
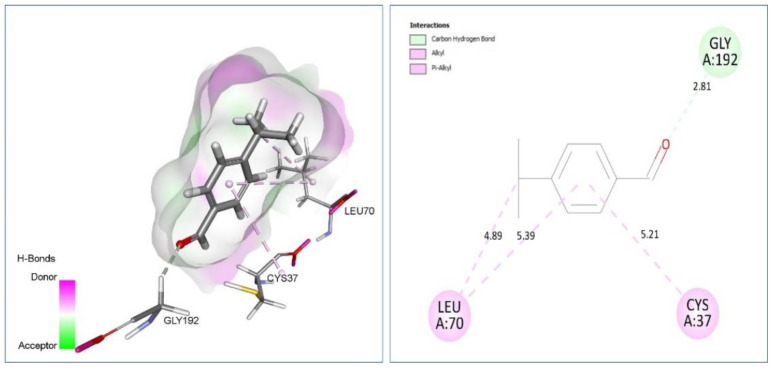
Two-dimensional (2D) and three-dimensional (3D) docking pose of cuminaldehyde in active site of tyrosyl-tRNA synthetase (PDB Id: 1JIJ) enzyme.

**Figure 5 plants-11-02236-f005:**
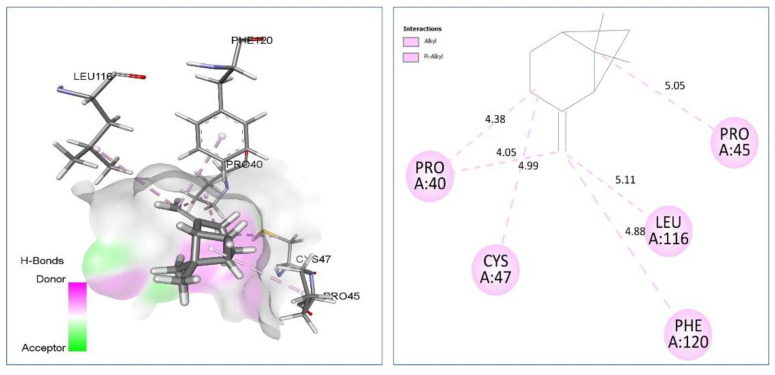
Two-dimensional (2D) and three-dimensional (3D) docking pose of β-pinene in active site of Human PRDX5 antioxidant enzyme (PDB ID, 1HD2).

**Figure 6 plants-11-02236-f006:**
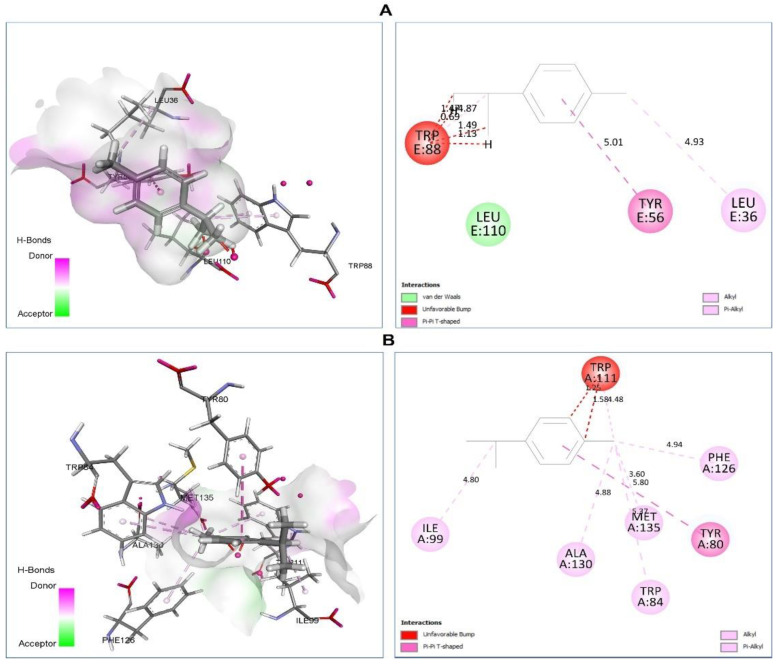
Two-dimensional (2D) and three-dimensional (3D) docking pose of *p*-cymene in active site of LasR (**A**) and CviR (**B**) enzymes.

**Table 1 plants-11-02236-t001:** Chemical composition of *C. cyminum* L. (seeds) EO assessed by GC/MS technique. ^a^: Linear Retention Index.

Code	Components	l.r.i. ^a^	Percentage	Molecular Weight	Chemical Formula
**1**	α-thujene	933	0.4	136.23	C_10_H_16_
**2**	α-pinene	941	0.9	136.23	C_10_H_16_
**3**	Sabinene	978	0.3	136.23	C_10_H_16_
**4**	β-**pinene**	**982**	**15.1**	136.238	C_10_H_16_
**5**	Myrcene	993	0.6	136.238	C_10_H_16_
**6**	α-phellandrene	1006	0.3	136.23	C_10_H_16_
**7**	** *p* ** **-cymene**	**1028**	**14.2**	134.22	C_10_H_14_
**8**	Limonene	1032	0.5	136.24	C_10_H_16_
**9**	**γ** **-terpinene**	**1064**	**14.4**	136.234	C_10_H_16_
**10**	Linalool	1101	0.1	154.253	C_10_H_18_O
**11**	4-terpineol	1179	0.4	154.25	C_10_H_18_O
**12**	α-terpineol	1191	0.2	154.25	C_10_H_18_O
**13**	**Cuminaldehyde**	**1240**	**42.4**	148.205	C_10_H_12_O
**14**	Carvone	1242	0.1	150.22	C_10_H_14_O
**15**	Phellandral	1274	0.2	152.23	C_10_H_16_O
**16**	**α-terpin-7-al**	**1283**	**5.2**	150.22	C_10_ H_14_O
**17**	γ-**terpin-7-al**	**1288**	**2.7**	150.22	C_10_H_14_O
**18**	β-caryophyllene	1419	0.3	204.36	C_15_H_24_
**19**	γ-muurolene	1478	0.4	204.35	C_15_H_24_
**20**	Carotol	1595	0.4	222.37	C_15_H_26_O
**Chemical classes**					
Monoterpene hydrocarbons			46.7%		
Oxygenated monoterpenes			51.3%		
Sesquiterpene hydrocarbons			0.7%		
Oxygenated sesquiterpenes			0.4%		
**Total identified**			**99.1%**		

**Table 2 plants-11-02236-t002:** Antioxidant activities of cumin EO. The letters (a–c) indicate a significant difference between the different antioxidant methods according to the Duncan test (*p* < 0.05).

Antioxidant Tests	Cumin EO IC_50_ (mg/mL)	AA EC_50_ (mg/mL)	BHT IC_50_ (mg/mL)	BHA IC_50_ (mg/mL)
**DPPH (IC_50_ mg/mL)**	8 ± 0.54 ^b^	12 ± 0.01 ^a^	11.50 ± 0.62 ^a^	-
**Reducing power (EC_50_ mg/mL)**	3.50 ± 0.03 ^c^	25 ± 0.01 ^a^	23.00 ± 1.0 ^b^	-
**β-carotene (IC_50_ mg/mL)**	3.80 ± 0.34 ^b^	-	4.60 ± 1.60 ^a^	-
**Chelating Power (IC_50_ mg/mL)**	8.40 ± 0.14 ^b^	-	-	32.50 ± 1.32 ^a^

**Table 3 plants-11-02236-t003:** Mean diameter of inhibition zone (mIZ ± mm), MICs, MBCs, and MBC/MIC ratio determination by disc diffusion and microdilution assays. The letters (a–k) indicate a significant difference between the different mZI according to the Duncan test (*p* < 0.05).

*Vibrio* spp. Tested	Cumin EO			
	mZI ± SD (mm)	MIC ± SD (mg/mL)	MBC ± SD (mg/mL)	MBC/MIC Ratio
*V. cholerae* ATCC 9459	34.33 ± 0.58 ^ef^	0.023	6	>4; Bacteriostatic
*V. vulnificus* ATCC 27562	30.33 ± 0.58 ^g^	0.023	1.5	>4; Bacteriostatic
*V. parahaemolyticus* ATCC 17802	39.67 ± 0.58 ^b^	0.046	12	>4; Bacteriostatic
*V. parahaemolyticus* ATCC 43996	28.67 ± 1.15 ^h^	0.023	1.5	>4; Bacteriostatic
*V. alginolyticus* ATCC 33787	34.67 ± 0.58 ^de^	0.023	3	>4; Bacteriostatic
*V. alginolyticus* ATCC 17749	33.33 ± 0.58 ^f^	0.023	6	>4; Bacteriostatic
*V. furnisii* ATCC 35016	11.33 ± 0.58 ^k^	0.023	3	>4; Bacteriostatic
*V. cincinnatiensis* ATCC 35912	14.67 ± 0.28 ^j^	0.046	12	>4; Bacteriostatic
*V. proteolyticus* ATCC 15338	30.33 ± 0.58 ^g^	0.023	6	>4; Bacteriostatic
*V. natrigens* ATCC 14048	36.67 ± 0.58 ^c^	0.023	3	>4; Bacteriostatic
*V. mimicus* ATCC 33653	28.67 ± 0.58 ^h^	0.046	12	>4; Bacteriostatic
*V. fluvialis* ATCC 33809	41.33 ± 1.15 ^a^	0.046	3	>4; Bacteriostatic
*V. carhiaccae* ATCC 35084	35.33 ± 0.58 ^d^	0.046	6	>4; Bacteriostatic
*V. harveyi* ATCC 18293	35.67 ± 0.58 ^cd^	0.023	3	>4; Bacteriostatic
*V. diazotrophicus* ATCC 33466	11.00 ± 0.00 ^k^	0.023	3	>4; Bacteriostatic
*V. tapetis* CECT 4600^T^	30.67 ± 0.58 ^g^	0.046	6	>4; Bacteriostatic
*V. splendidus* ATCC 33125	26.33 ± 0.58 ^i^	0.046	6	>4; Bacteriostatic

**Table 4 plants-11-02236-t004:** Swarming inhibition on Lauria Bertani (0.5% agar-agar) by cumin EO and cuminaldehyde. The letters (a–f) indicate a significant difference between the diameter of colony tested at different concentrations according to the Duncan test (*p* < 0.05).

	Control	Concentrations Tested (mg/mL)				
		10	50	125	250	500
**Diameter of the colony (mm** **± SD)**						
**Cumin EO**	54.00 ± 0.00 ^a^	19.33 ± 0.57 ^b^	14.67 ± 0.57 ^c^	12.00 ± 0.00 ^d^	10.33 ± 0.57 ^e^	8.67 ± 0.57 ^f^
**Cuminaldehyde**	54.00 ± 0.00 ^a^	15.67 ± 0.57 ^b^	13.67 ± 0.57 ^c^	12.00 ± 0.00 ^d^	10.33 ± 0.57 ^e^	9.00 ± 0.00 ^f^
**Percentage of motility inhibition (%)**						
**Cumin EO**	100 ± 0.00	64.20 ± 0.57	77.15 ± 0.57	84.45 ± 0.00	87.76 ± 0.57	90.12 ± 0.57
**Cuminaldehyde**	100 ± 0.00	70.99 ± 0.57	80.75 ± 0.57	85.96 ± 0.57	87.98 ± 0.57	89.77 ± 0.00

**Table 5 plants-11-02236-t005:** ADMET properties of compounds the major phytocompounds. Number of the compounds are same listed in Table 1.

Entry	4	7	9	13	16	17	Reference
**Absorption**							
Water solubility	−4.221	−5.163	−3.941	−3.923	−2.79	−2.79	-
Caco2 permeability	1.373	1.399	1.414	1.503	1.517	1.517	>0.9
Intestinal absorption (human)	94.607	94.256	96.219	95.543	97.506	97.506	<30% is poorly
Skin Permeability (log Kp)	−1.646	−1.2	−1.489	−1.425	−2.624	−2.624	>−2.5 is low
**Distribution**							
P-glycoprotein substrate	No	No	No	No	No	No	No
P-glycoprotein I inhibitor	No	No	No	No	No	No	No
P-glycoprotein II inhibitor	No	No	No	No	No	No	No
VDss (human)	0.68	0.455	0.412	0.274	0.233	0.233	Low is <−0.15, High is >0.45
Fraction unbound (human)	0.353	0.262	0.42	0.305	0.465	0.465	-
BBB permeability	0.812	0.785	0.754	0.664	0.633	0.633	Poorly is <−1, High is >0.3
CNS permeability	−1.837	−1.359	−2.049	−1.506	−2.197	−2.197	Penetrate is >−2, Unable is <−3
**Metabolism**							
CYP2D6 substrate	No	No	No	No	No	No	No
CYP3A4 substrate	No	No	No	No	No	No	-
CYP1A2 inhibitior	No	No	No	No	No	No	No
CYP2C19 inhibitior	No	No	No	No	No	No	No
CYP2C9 inhibitior	No	No	No	No	No	No	No
CYP2D6 inhibitior	No	No	No	No	No	No	No
CYP3A4 inhibitior	No	No	No	No	No	No	No
**Excretion**							
Total Clearance	0.03	1.163	0.217	0.212	0.182	0.182	-
Renal OCT2 substrate	No	No	No	No	No	No	-
**Toxicity**							
AMES toxicity	No	No	No	No	No	No	No
Max. tolerated dose (human)	0.24	0.193	0.756	0.128	0.723	0.723	Low is ≤0.477, High is >0.477
hERG I inhibitor	No	No	No	No	No	No	No
hERG II inhibitor	No	No	No	No	No	No	No
Oral Rat Acute Toxicity (LD50)	1.617	1.533	1.766	1.499	1.971	1.971	-
Oral Rat Chronic Toxicity (LOAEL)	2.247	2.411	2.394	2.052	2.034	2.034	-
Hepatotoxicity	No	No	No	No	No	No	No
Skin Sensitisation	No	No	No	Yes	Yes	Yes	No
*T.Pyriformis* toxicity	0.633	0.767	0.627	0.765	0.732	0.732	>−0.5 is toxic
Minnow toxicity	1.131	0.65	0.906	0.862	1.118	1.118	<−0.3 is toxic

**Table 6 plants-11-02236-t006:** Best phytoconstituents identified from *C. cyminum* L. EO with the lowest binding energies and their interaction residues with selected target proteins.

Compounds	Interacting Residues Receptor vs. Targets	Binding Energy (kcal/mol)
β-pinene vs. 1HD2	**Alkyl/Pi-Alkyl:** Pro40 (4.05) (4.38), Pro45 (5.05), Cys47 (4.99), Leu116 (5.11), Phe120 (4.88).	−4.6
Cuminaldehyde vs. 1HD2	**H bond:** Thr147 (2.10). **Alkyl:** Pro45 (5.14), Cys47 (5.00).	−5.4
Cuminaldehyde vs. 1JIJ	**C-H bond:** Gly192 (2.81). **Alkyl/Pi-Alkyl:** Cys37 (5.21), Leu70 (4.89) (5.39).	−7.4
β-Caryophyllene vs. IJIJ	**Alkyl/Pi-Alkyl:** Cys37(4.64), Ala39 (4.28) (4.53) (4.75), Pro53 (5.37) (4.50), His50 (4.00) (5.06).	−6.4
*p*-Cymene vs. 2UV0	**van der Waals:** Leu110. **Unfavorable Bump:** Trp88 (0.69) (1.13) (1.47) (1.49). **Pi-Pi T-Shaped:** Tyr56 (5.01). **Alkyl/Pi-Alkyl:** Leu36 (4.93), Trp88 (4.87).	−7.4
γ-Terpinene vs. 2UV0	**Unfavorable Bump:** Trp88 (0.61) (1.34) (1.54). **Alkyl/Pi-Alkyl:** Leu36 (4.93), Tyr56 (5.20), Tyr64 (3.76) (4.88), Trp88 (4.67).	−7.4
Cuminaldehyde vs. 2UV0	**H bond:** Arg61 (4.31). **Pi-Lone Pair:** Tyr64 (2.79). **Unfavorable Bump:** Trp88 (0.71) (0.75) (0.22) (1.39). **Pi-Pi T-Shaped:** Tyr56 (4.90). **Alkyl/Pi-Alkyl:** Trp88 (4.75).	−7.4
*p*-Cymene vs. 3QP1	**Unfavorable Bump:** Trp111 (1.25) (1.58). **Pi-Pi T-Shaped:** Tyr80 (5.80). **Alkyl/Pi-Alkyl:** Trp84 (3.37) Ile99 (4.80), Phe126m(4.94), Ala130 (4.88). Met135 (5.80).	−7.5
γ-Terpinene vs. 3QP1	**van der Waals:** Leu57, trp84, Tyr88, Ile99, Leu100. **Unfavorable Bump:** Trp111 (0.90) (1.41). **Alkyl/Pi-Alkyl:** Tyr80 (5.36), Phe115 (4.99), Phe126 (4.99), Ala130 (5.01), Met135 (3.80) (5.35), Trp111 (4.20) (4.45).	−7.5
Cuminaldehyde vs. 3QP1	**Pi-Pi T-Shaped:** Tyr80 (5.82). **Pi-Alkyl:** Ile99 (4.76).	−7.2

**Table 7 plants-11-02236-t007:** Review of the chemical composition of *C. cyminum* EO from seeds.

Origin	Chemical Composition (Main Constituents)	References
**China**	Cuminaldehyde (36.31%), cuminic alcohol (16.92%), γ-terpinene (11.14%), safranal (10.87%), *p*-cymene (9.85%) and β-pinene (7.75%)	[31]
**Iran**	α-pinene (29.1%), limonene (21.5%), 1,8-cineole (17.9%), and linalool (10.4%)	[32]
	Cuminaldehyde (25.2%), p-mentha-1,3-dien-7-al (13%), p-mentha-1,4-dien-7-al (16.6%), γ-terpinene (19%), p-cymene (7.2%), and β-pinene (10.4%).	[33]
	α-Pinene (29.2%), limonene (21.7%), 1,8-cineole (18.1%), linalool (10.5%), linalyl acetate (4.8%), and α-terpineole (3.17%).	[34]
	α-pinene (30.12%), limonene (10.11%), 1,8-cineole (11.54%), γ-terpinene (3.56%), linalool (10.3%), sabinene (1.11%), *p*-cymene (0.6%), α-campholenal (1.76%), linalyl acetate (4.76%), α-terpinyl acetate (1.8%), neryl acetate (1%).	[35]
	Cuminaldehyde (28.24%), γ-terpinene (21.39%), o-Cymene (13.78%), β-pinene (3.14%), and β-Acoradiene (1.68%).	[36]
	3-caren-10-al (47.27%), cuminal (25.92%), 2-caren-10-al (8.05%), γ-terpinene (7.66%), (-)-β-pinene (5.11%), and *p*-cymene (2.71%).	[37]
	Cuminaldehyde (38.26%), α,β-dihydroxy ethylbenzene (29.16%), 2-caren-10-al (11.20%), γ-terpinene (6.49%), and β-pinene (5.25%).	[38]
	Cuminaldehyde (29.0%), α-terpinen-7-al (20.7%), γ-terpinene (12.94%), γ-terpinen-7-al (8.91%), *p*-cymene (8.55%), and β-pinene (7.72%).	[39]
**India**	Safranal (16.8–29.0%), γ-terpinene (14.1–19.6%), γ-terpinene-7-al (13.5–25.5%), cuminaldehyde (17.5–22.3%), β-pinene (6.8–10.4%), and *p*-cymene (4.1–8.8%).	[40]
	Cuminaldehyde (49.4%), *p*-cymene (17.4%), β-pinene (6.3%), α-terpinen-7-al (6.8%), γ-terpinene (6.1%), *p*-cymen-7-ol (4.6%), and thymol (2.8%).	[41]
	Cuminaldehyde (36.67%), caren-10-al (21.34%), β-pinene (18.76%), γ-terpinene (16.86%), terpinen-4-ol (2.44%), α-thujene (1.88%), α-pinene (1.41%), *p*-cymene (0.30%), carbicol (0.19%) and α-terpineol (0.09%).	[42]
**China**	Cuminaldehyde (44.53%), *p*-cymene (12.14%), β-pinene (10.47%) and γ-terpinene (8.40%)	[43]
**Thailand**	Cumin aldehyde (33.94%), α-terpinen-7-al (32.20%), γ-terpinen-7-al (13.74%), γ-terpinene (6.67%), β-pinene (5.34%) and *p*-cymene (3.58%).	[44]
	Cuminaldehyde (27.10%), β-pinene (25.04%) and γ-terpinene (15.68%).	[45]
**Tunisia**	γ-terpinen (25.58%), 1-phenyl-1,2 ethanediol (23.16%), cuminaldehyde (15.31%), β-pinene (15.16%), and ρ-cymene (9.05%)	[46]
	Cuminaldehyde (39.48%), γ-terpinene (15.21%), O-cymene (11.82%), β-pinene (11.13%), 2-caren-10-al 7.93%), trans-carveol (4.49%) and myrtenal (3.5%).	[47]
	Cuminaldehyde (28.22%), 1-phenyl-1-butanol (23.33%), β-pinene (12.61%) and *p*-cymene (11.72%).	[48]
**Sudan**	2-Caren-10-al (29.64%), benzaldehyde, 4-1-methyethyl (16.58%), and 2-J-pinene (12.06%)	[49]
**Spain**	Cuminaldehyde (34.11%), Δ2-Caren-10-al (20.78%), *p*-cymene (12.25%), Δ3-C10-al (11.80%), Δ4-Carene (10.47%), β-pinene (7.3%).	[50]
**Iran**	Cuminaldehyde (41.5%), *p*-cymene (17.4%), β-pinene (10.7%), γ-Terpinene (6.5%), *p*-mentha-1,3-dien-7-al (5.5%), *p*-mentha-1,4-dien-7-al (1.5%), β-acoradiene (3.5%).	[51]
**Egypt**	Cuminaldehyde (29.3%), γ-Terpinene (18.5%), β-pinene (15.7%), *p*-mentha-1,3-dien-7-al (10.6%), *p*-cymene (10.1%), *p*-mentha-1,4-dien-7-al (7.6%), β-acoradiene (0.2%).	
**India**	γ-Terpinene (31.1%), cuminaldehyde (23.2%), *p*-cymene (18.4%), β-pinene (12.6%), *p*-mentha-1,3-dien-7-al (7.2%), *p*-mentha-1,4-dien-7-al (0.4%), β-acoradiene (0.1%).	
**Europe**	γ-Terpinene (26.5%), cuminaldehyde (22.4%), *p*-cymene (20.2%), β-pinene (14.1%), *p*-mentha-1,3-dien-7-al (6.6%), *p*-mentha-1,4-dien-7-al (1.4%), β-acoradiene (0.3%).	
**Morocco**	β-pinene (20.8–86.4%), *p*-cymene (6.2–24.7%), γ-terpinene (18.1–90.7%), cuminaldehyde (51.5–91.5%), α-terpinen-7-al (21.2–95.3%) and α-terpinen-7-al (22.6–55.06%)	[52]

**Table 8 plants-11-02236-t008:** Review of the antibacterial activities of some EO against *Vibrio* species.

Plant Species Tested	*Vibrio* Species Tested	References
*Bauhinia variegata*	*V. cholerae*	[54]
*Psidium guajava*, *Azadirachta indica*	*V. cholerae*	[55]
*Mentha pulegium*	*V. cholerae*	[56]
*Syzygium aromaticum*	*V. parahaemolyticus*	[57]
*Mentha longifolia*; *M. pulegium*; *Eugenia caryophyllata*; *Rosmarinus officinalis and Thymus vulgaris*	*V. alginolyticus*, *V. parahaemolyticus*, *V. fluvialis*, *V. vulnificus*	[58]
*C. cyminum*	*V. cholerae*, *V. parahaemolyticus*, *V. alginolyticus*, *V. vulnificus*, *V. harveyi*, *V. proteolyticus*, *V. furnisii*, *V. mimicus*, *V. furnisii*, *V. natrigens*, *V. carhiaccae*, *V. fluvialis*	[47]
*Ocimum basilicum*	*V. parahaemolyticus*, *V. mimicus*	[59]
*Satureja bachtiarica Bunge*, *Zataria multiflora*	*V. parahaemolyticus*, *V. harveyi.*	[60]
*Cymbopogon nardus*	*V. damsela*, *Vibrio spp.*	[61]
*Lippia berlandieri*	*V. cholerae*, *V. parahaemolyticus*, *V. vulnificus*	[62]
*Cordia globosa*	*V. cholerae*	[63]
*Eucalyptus globulus*	*V. cholerae*	[64]
	*V. harveyi*, *V. ichthyoenteri*	[65]
*Mentha piperita*	*V. parahaemolyticus*, *V. cholerae*, *V. vulnificus*, *V. alginolyticus*, *V. mimicus*, *V. damsela*, *V. campbellii*, *V. harveyi*, *V. logei*	[66]
*Elettaria cardamomum*, *Mentha spicata*, *Petroselinum crispum*, *Ocimum basilicum*	*V. cholerae*, *V. vulnificus*, *V. parahaemolyticus*, *V. alginolyticus*, *V. furnisii*, *V. cincinnatiensis*, *V. proteolyticus*, *V. natrigens*, *V. mimicus*, *V. fluvialis*, *V. anguillarum*, *V. carrichariae*, *V. harveyii*, *V. diazotrophicus*, *V. tapetis*, *V. splendidus.*	[67,68,69]
*Nigella sativa*	*V. parahaemolyticus*	[70]
*Origanum majorana*	*V. parahaemolyticus*, *V. alginolyticus*	[21]
*Artemisia absinthium*, *Zataria multiflora*, *Pulicaria gnaphalodes*, *Trachyspermum ammi*, *Cuminum cyminum*	*V. parahaemolyticus*	[71]
*Alpinia galanga*, *Zingiber officinale*	*V. cholerae*	[72]
*Origannum majorana*, *Cinnamomum verum*	*V. parahaemolyticus*, *V. cholerae*	[73]
*Protium heptaphyllum*	*V. parahaemolyticus*	[74]
*Abies alba*, *Apium graveolens*, *Artemisia dracunculus*, *A. herba alba*, *Cinnamomum camphora*, *C. cassia*, *C. zeylanicum*, *Citrus sinensis*, *C. cyminum*, *Curcuma longa*, *Cymbopogon martini*, *E. citriodora*, *E. dives*, *Laurus nobilis*, *Litsea citrata*, *Melaleuca alternifolia*, *Mentha × piperita*, *M. pulegium*, *P. crispum*, *Pogostemon cablin*, *Thymus zygis*, *Zingiber officinalis.*	*V. campbellii*, *V. parahaemolyticus*	[75]
Clove, thyme, garlic	*V. parahaemolyticus*	[76]
*Carum carvi*, *Coriandrum sativum L.*	*V. parahaemolyticus*, *V. alginolyticus*, *V. proteolyticus*, *V. furnisii*, *V. mimicus*, *V. natrigens*, *V. carhiaccae*, *V. fluvialis*	[77]
*Carum carvi*	*V. cholerae*, *V. vulnificus*, *V. parahaemolyticus*, *V. alginolyticus*, *V. furnisii*, *V. cincinnatiensis*, *V. proteolyticus*, *V. natrigens*, *V. mimicus*, *V. fluvialis*, *V. anguillarum*, *V. carrichariae*, *V. harveyii*, *V. diazotrophicus*, *V. tapetis*, *V. splendidus.*	[53]

## Data Availability

The data generated and analyzed during this study are included in this article.

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
