# Peer review of "In Vitro and In Silico Screening of Anti-Vibrio spp., Antibiofilm, Antioxidant and Anti-Quorum Sensing Activities of Cuminum cyminum L. Volatile Oil"

_plants, 2022, doi:10.3390/plants11172236_

Round 1
Reviewer 1 Report
The authors of the manuscript “In vitro and in silico screening of anti-Vibrio spp., antibiofilm, antioxidant and anti-quorum sensing activities of Cuminum cyminum L. volatile oil (Chemovar Cuminaldehyde)” present an interesting work based on chemical composition, antioxidant and vibriocidal activities of essential oil extracted from Cumin. Cumin EO was also tested on the abilities to inhibit the biofilm formation and secretion of quorum sensing controlled virulence traits in Chromobacterium violaceum and Pseudomonas aeruginosa strains were also reported.
The manuscript is clearly written and the results are interesting but contains some flaws that need to be improved before acceptance.
Lines 21, 22: Please, authors should write the name of species in italic format.
Lines 25, 26: The cited bacteria species are first time written. Please, use the regular full binomial name (e.g. Vibrio fluvialis).
Lines 32, 33: See lines 25, 26.
Line 36: Please, use italic format on “P. aeruginosa”.
Line 59: Authors wrote “Pseudomonas aeruginosa”. Please use italic format, and it should be written P. aeruginosa. It has been already cited.
Lines 60 and 62: Please, use italic format on “P. aeruginosa”.
Line 70: about references, there is a possible wrong number sequence, please check number 21. Probably authors have to put 20 before, and then 21.
Line 125 Table 1: Please, insert a new table number on "chemical classes" (e.g. Table 2)
Line 155: Please, correct milliliter. Authors wrote “Ml” instead of mL.
Line 161 Table 3: Authors should better detail the legend of table 3 regarding the letters on values. They shoul give informations about test and statistic used.
Line 188: Title of paragraph should be written in italic format.
Line 215: Please, Authors should write Figure 1 instead Figure 2.
Line 217: Please, specify if it is a triplicate or other.
Line 219: See line 188.
Line 265: The figure 3 is not cited on text. Authors: a) they may improve the figure inserting curves and filling the legend with informations about statistics, and insert the citation on text; b) delete the figure.
Line 277 Table 4: See Table 3.
Line 318 Figure 4: The figure shows same both y axes title but different values, no panel indications (A, B). The legend of the figure is not complete (where is panel B) and lack of statistics. Please, Authors have to correct figure and legend.
Line 321: See line 188.
Line 368: See line 188.
Line 374: “S. aureus” is cited for first time. It should be written as “Staphylococcus aureus”.
Line 379: “S. aureus”, it should be written in italic format
Line 503: “E. coli” should be written as “Escherichia coli”, see line 374
Line 505: “C. tropicalis, C. glabrata” should be written as “Candida tropicalis, Candida glabrata”, see line 374
Table 8: “P. Crispum” should be written as full binomial name.
Line 525: “V. parahaemolyticus” should be written in italic format.
Line 528: “K. Pneumoniae” should be written as full binomial name.
4. Material and Methods: All paragraph titles should be written in italic format.
Line 585: “Pseudomonas aeruginosa” should be written as “P. aeruginosa”.
Line 594, 595: There are some words to check (e.g. mol-ecules, Au-toGrid)
Conclusion: Authors wrote “These results highlighted the potential use of this culinary spices as source of compounds able to prevent food contamination by Vibrio species”. This is a strong information. With the aim to support this information, authors should briefly discuss the comparison of MIC/MBC between EOs and an antibiotic or preservative commonly used in clinic research or food industries, and discuss the real toxicity of EO on animal cells (data available on general bibliography).
Author Response
Dear referee
Special thanks for your comments.
All modifications were highlighted in yellow in the new version of our manuscript
Special thanks
Review Report Form
Open Review
(x) I would not like to sign my review report
( ) I would like to sign my review report
English language and style
( ) Extensive editing of English language and style required
( ) Moderate English changes required
( ) English language and style are fine/minor spell check required
(x) I don't feel qualified to judge about the English language and style
Yes |
Can be improved |
Must be improved |
|
Does the introduction provide sufficient background and include all relevant references? |
(x) |
( ) |
( ) |
Are all the cited references relevant to the research? |
(x) |
( ) |
( ) |
Is the research design appropriate? |
(x) |
( ) |
( ) |
Are the methods adequately described? |
( ) |
(x) |
( ) |
Are the results clearly presented? |
( ) |
(x) |
( ) |
Are the conclusions supported by the results? |
( ) |
(x) |
( ) |
Comments and Suggestions for Authors
The authors of the manuscript “In vitro and in silico screening of anti-Vibrio spp., antibiofilm, antioxidant and anti-quorum sensing activities of Cuminum cyminum L. volatile oil (Chemovar Cuminaldehyde)” present an interesting work based on chemical composition, antioxidant and vibriocidal activities of essential oil extracted from Cumin. Cumin EO was also tested on the abilities to inhibit the biofilm formation and secretion of quorum sensing controlled virulence traits in Chromobacterium violaceum and Pseudomonas aeruginosa strains were also reported.
The manuscript is clearly written and the results are interesting but contains some flaws that need to be improved before acceptance.
Lines 21, 22: Please, authors should write the name of species in italic format.
Response: Modified as suggested.
Lines 25, 26: The cited bacteria species are first time written. Please, use the regular full binomial name (e.g. Vibrio fluvialis).
Response: Modified as suggested.
Lines 32, 33: See lines 25, 26.
Response: Modified as suggested.
Line 36: Please, use italic format on “P. aeruginosa”.
Response: Modified as suggested.
Line 59: Authors wrote “Pseudomonas aeruginosa”. Please use italic format, and it should be written P. aeruginosa. It has been already cited.
Response: Modified as suggested.
Lines 60 and 62: Please, use italic format on “P. aeruginosa”.
Response: Modified as suggested.
Line 70: about references, there is a possible wrong number sequence, please check number 21. Probably authors have to put 20 before, and then 21.
Response: Modified.
Line 125 Table 1: Please, insert a new table number on "chemical classes" (e.g. Table 2)
Response: We have used the same table format to present the chemical composition of essential oils were chemical classes are listed in the end of the table.
Line 155: Please, correct milliliter. Authors wrote “Ml” instead of mL.
Response: Modified as suggested.
Line 161 Table 3: Authors should better detail the legend of table 3 regarding the letters on values. They shoul give informations about test and statistic used.
Response: Modified as suggested. Information about the statistical test used were added.
Line 188: Title of paragraph should be written in italic format.
Response: Modified as suggested.
Line 215: Please, Authors should write Figure 1 instead Figure 2.
Response: Modified as suggested.
Line 217: Please, specify if it is a triplicate or other.
Response: Modified as suggested.
Line 219: See line 188.
Response: Modified as suggested.
Line 265: The figure 3 is not cited on text. Authors: a) they may improve the figure inserting curves and filling the legend with information about statistics, and insert the citation on text; b) delete the figure.
Response: Modified as suggested. This curve was deleted.
Line 277 Table 4: See Table 3.
Response: Modified as suggested.
Line 318 Figure 4: The figure shows same both y axes title but different values, no panel indications (A, B). The legend of the figure is not complete (where is panel B) and lack of statistics. Please, Authors have to correct figure and legend.
Response: Modified as suggested.
Figure 3. Inhibition of the proteolytic activity (A) and elastolytic activity (B) in P. aeruginosa PAO1 strain by different concentration of Cumin EO and cuminaldehyde. Values are the average of at least three independent determinations. Means followed by the same letters are not significantly different at p<0.05 based on Duncan’s multiple range test.
Line 321: See line 188.
Response: Modified as suggested.
Line 368: See line 188.
Response: Modified as suggested.
Line 374: “S. aureus” is cited for first time. It should be written as “Staphylococcus aureus”.
Response: Modified as suggested.
Line 379: “S. aureus”, it should be written in italic format.
Response: Modified as suggested.
Line 503: “E. coli” should be written as “Escherichia coli”, see line 374
Response: Modified as suggested.
Line 505: “C. tropicalis, C. glabrata” should be written as “Candida tropicalis, Candida glabrata”, see line 374
Response: Modified as suggested.
Table 8: “P. Crispum” should be written as full binomial name.
Response: Modified as suggested.
Line 525: “V. parahaemolyticus” should be written in italic format.
Response: Modified as suggested.
Line 528: “K. Pneumoniae” should be written as full binomial name.
Response: Modified as suggested.
- Material and Methods:All paragraph titles should be written in italic format.
Response: Modified as suggested.
Line 585: “Pseudomonas aeruginosa” should be written as “P. aeruginosa”.
Response: Modified as suggested.
Line 594, 595: There are some words to check (e.g. mol-ecules, Au-toGrid)
Response: Modified as suggested.
Conclusion: Authors wrote “These results highlighted the potential use of this culinary spices as source of compounds able to prevent food contamination by Vibrio species”. This is a strong information. With the aim to support this information, authors should briefly discuss the comparison of MIC/MBC between EOs and an antibiotic or preservative commonly used in clinic research or food industries, and discuss the real toxicity of EO on animal cells (data available on general bibliography).
Response: Modified as suggested.
In summary, our results indicated that cuminaldehyde, β-pinene, γ-terpinene, and p-cymene, were the main phytoconstituents identified in cumin essential oil by GC/MS technique. This chemovar was particularly active against planktonic and biofilm forming V. alginolyticus, V. cholerae, V. vulnificus, and V. parahaemolyticus species. The same essential oil and its main compound (cuminaldehyde) were able to modulate the expression of violacein production in C. violaceum in a concentration dependent manner. At low concentrations, cumin essential oil and cuminaldehyde were able to inhibit the flagellar motility of P. aeruginosa PAO1 strain and attenuate the production of elastase and protease. Further analyses are necessary to elucidate the mechanism of action of Cumin EO and its role to prevent seafood product contamination by spoilage bacterial belonging to Vibrio genus.
Submission Date
31 July 2022
Date of this review
05 Aug 2022 14:20:58

Reviewer 2 Report
Dear authors, I suggest that you reconsider the main focus of the manuscript. It seems very confusing and not well designed. Although you included many techniques somehow they seem very disintegrated. Molecular docking analysis includes testing of compounds without scientific novelty. Considering that your focus was on cuminaldehyde, it should be kept throughout the study. Also I do not see the reasoning for testing on human antioxidant enzyme. EC50 values ​​for antioxidant tests also seem very high. Please not that there are many mistakes and inconsistencies in writing which need to be corrected. I made some remarks directly in the manuscript.
Regards

Author Response
Dear referee
Special thanks for your comments.
All modifications were highlighted in yellow in the new version of our manuscript
Special thanks
Review Report Form
Open Review
( ) I would not like to sign my review report
(x) I would like to sign my review report
English language and style
( ) Extensive editing of English language and style required
( ) Moderate English changes required
( ) English language and style are fine/minor spell check required
(x) I don't feel qualified to judge about the English language and style
Yes |
Can be improved |
Must be improved |
|
Does the introduction provide sufficient background and include all relevant references? |
( ) |
(x) |
( ) |
Are all the cited references relevant to the research? |
(x) |
( ) |
( ) |
Is the research design appropriate? |
( ) |
( ) |
(x) |
Are the methods adequately described? |
( ) |
(x) |
( ) |
Are the results clearly presented? |
( ) |
( ) |
(x) |
Are the conclusions supported by the results? |
( ) |
( ) |
(x) |
Comments and Suggestions for Authors
Dear authors, I suggest that you reconsider the main focus of the manuscript. It seems very confusing and not well designed. Although you included many techniques somehow, they seem very disintegrated. Molecular docking analysis includes testing of compounds without scientific novelty. Considering that your focus was on cuminaldehyde, it should be kept throughout the study. Also, I do not see the reasoning for testing on human antioxidant enzyme. EC50 values ​​for antioxidant tests also seem very high. Please not that there are many mistakes and inconsistencies in writing which need to be corrected. I made some remarks directly in the manuscript.
I suggest that you reconsider the main focus of the manuscript. It seems very confusing and not well designed. Although you included many techniques somehow they seem very disintegrated.
Response: Dear colleague: Special thanks for your valuable remark. In fact, whole manuscript was deeply revised and reorganized. Abstract was rewritten and reorganized. Results were rewritten and some statistical analysis tests were added as suggested by referees. Material and methods section was detailed, and conclusion part was rechecked.
The main aims were clearly mentioned: Hence, in view of the attributed medicinal significance of the cumin plant and its availability as medicinally resource, the present work focuses specifically on an aromatic plant commonly used in the Saudi kitchen to prepare fish and shellfish dishes. We aimed to explore the constituents of its EO and to evaluate in vitro, its anti-Vibrio activities. The ability of the obtained Cumin EO to scavenge reactive oxygen species using different as-says was also assessed. Moreover, computational study has been achieved to elucidate the physicochemical properties, pharmacokinetic properties, druglikeness, and toxicity pre-diction of the main bioactive compounds from Cumin EO. To get insight into the interaction mode of these bioactive molecules with known target enzymes involved in antioxidant, antibacterial, and anti-quorum sensing activities, a molecular docking approach was performed.
Molecular docking analysis includes testing of compounds without scientific novelty.
Response: This work is original in terms of tested receptors and activities. Therefore, docking analysis have been successful applications that demonstrated how molecular docking is becoming a powerful tool in the discovery of drug candidates via testing the main targets. Additionally, molecular docking is a method which analyses the conformation and orientation (referred together as the “pose”) of molecules into the binding site of a macromolecular target. It has been established as a pivotal technique among the computational tools for structure-based drug discovery attempts to predict how a small molecule (the ligand) binds to a protein receptor (the target) by simulating the physical interaction between the two. Below, we summarize some of the advantages of molecular docking over simple physicochemical properties (e.g., logP) with regard to benchmarking:
1- Interpretability: docking scores have a structural interpretation in terms of predicted binding affinity, correlating with experimental values in some protein families.
2- Relevance: docking scores are routinely employed by medicinal chemists in academia and industry to discover hits in virtual screening experiments. Docking poses are also used to identify and exploit important interactions during lead optimization.
3- Computational cost: docking scores can typically be computed in under a minute, unlike other computational methods like free energy perturbation calculations or density functional theory.
4- Challenging benchmark: the relationship between molecular structure and docking score is complex, as the docking score depends on the 3D structure of the ligand–target complex.
Considering that your focus was on cuminaldehyde, it should be kept throughout the study. Also I do not see the reasoning for testing on human antioxidant enzyme.
Response: Concerning the enzyme Human Peroxiredoxin 5, known as a novel thioredoxin peroxidase, which was widely expressed in mammalian tissues suggesting its important role as antioxidant in organelles that are major sources of ROS, has been selected as a potent antioxidant target.
EC50 values ​​for antioxidant tests also seem very high.
As given by the experimenter and was confirmed by the literature survey.
https://doi.org/10.1016/j.sajb.2009.10.009
Please not that there are many mistakes and inconsistencies in writing which need to be corrected. I made some remarks directly in the manuscript.
Response: Checked as suggested.
Regards
Submission Date
31 July 2022
Date of this review
08 Aug 2022 19:34:01

Round 2
Reviewer 2 Report
Thank you for extensive response!